# Influence of the Shod Condition on Running Power Output: An Analysis in Recreationally Active Endurance Runners

**DOI:** 10.3390/s22134828

**Published:** 2022-06-26

**Authors:** Diego Jaén-Carrillo, Luis E. Roche-Seruendo, Alejandro Molina-Molina, Silvia Cardiel-Sánchez, Antonio Cartón-Llorente, Felipe García-Pinillos

**Affiliations:** 1Campus Universitario, Universidad San Jorge, Autov A23 km 299, Villanueva de Gállego, 50830 Zaragoza, Spain; djaen@usj.es (D.J.-C.); leroche@usj.es (L.E.R.-S.); scardiels@usj.es (S.C.-S.); acarton@usj.es (A.C.-L.); 2Department of Physical Education and Sport, University of Granada, 18016 Granada, Spain; fegarpi@gmail.com; 3Departamento de Educación Física, Deporte y Recreación, Universidad de La Frontera, Temuco 4780000, Chile

**Keywords:** barefoot, footstrike, stiffness, sensor, wearable

## Abstract

Several studies have already analysed power output in running or the relation between VO2max and power production as factors related to running economy; however, there are no studies assessing the difference in power output between shod and barefoot running. This study aims to identify the effect of footwear on the power output endurance runner. Forty-one endurance runners (16 female) were evaluated at shod and barefoot running over a one-session running protocol at their preferred comfortable velocity (11.71 ± 1.07 km·h^−1^). The mean power output (MPO) and normalized MPO (MPOnorm), form power, vertical oscillation, leg stiffness, running effectiveness and spatiotemporal parameters were obtained using the Stryd™ foot pod system. Additionally, footstrike patterns were measured using high-speed video at 240 Hz. No differences were noted in MPO (*p* = 0.582) and MPOnorm (*p* = 0.568), whereas significant differences were found in form power, in both absolute (*p* = 0.001) and relative values (*p* < 0.001), running effectiveness (*p* = 0.006), stiffness (*p* = 0.002) and vertical oscillation (*p* < 0.001). By running barefoot, lower values for contact time (*p* < 0.001) and step length (*p* = 0.003) were obtained with greater step frequency (*p* < 0.001), compared to shod running. The prevalence of footstrike pattern significantly differs between conditions, with 19.5% of runners showing a rearfoot strike, whereas no runners showed a rearfoot strike during barefoot running. Running barefoot showed greater running effectiveness in comparison with shod running, and was consistent with lower values in form power and lower vertical oscillation. From a practical perspective, the long-term effect of barefoot running drills might lead to increased running efficiency and leg stiffness in endurance runners, affecting running economy.

## 1. Introduction

Endurance running events range from 3000 m to over 160 km in ultra-marathons. Nowadays, both the number of runners in endurance races and the number of organised races have increased. For example, 19,076 runners (19.51% women) finished the half marathon in Valencia in 2020 (Spain). The lower limb muscles execute three distinctive functions during such events: (i) force and power generation; (ii) shock absorption; and (iii) store and release elastic energy [1], thus compromising running economy. Endurance runners experienced improvements in muscle strength and power, among others, after an 8-week training intervention directly affecting running economy and, thus, performance [2]. The novel appearance of wearable devices capable of obtaining kinetic and kinematic data during running offers sports practitioners a new way to quantify workload by acquiring valuable metrics such as spatiotemporal parameters, power, and leg stiffness.

Although a wide range of endurance runners tend to collide with the ground first with the heel when shod [3], the switch from shod to barefoot running implies a tendency toward a midfoot (MFS) or forefoot strike pattern (FFS), influencing factors such as contact (CT) and flight time (FT), step frequency (SF), step length (SL), loading rate and leg compliance [4,5,6,7]. While leg stiffness increases with barefoot running in comparison to shod running [8], a significant reduction in running dynamic stability was found when changing from shod to barefoot conditions [9]. Moreover, greater activation levels in the intrinsic muscles of the foot have been found with the stance phase in barefoot running, in comparison with shod running, producing adjustment during the compression of the longitudinal arch. This results in a greater ability to recoil elastic energy when running barefoot [10,11].

The recent appearance of wearable power meters on the running scene may change training and competition by providing power output values for endurance runners. These sensors may allow us to monitor and quantify workload from a fair and objective perspective with accurate replication, as they already do in cycling [12]. Velocity and both the body height and weight of a runner, as well as external conditions such as slope and wind, may influence power output in running [13,14]. Although the level of agreement between power meter systems in running and two theoretical models for power output analysis has been assessed [15], the lack of scientific evidence for the use and interpretation of such metrics in endurance runners may prevent sport practitioners from adopting them as a means to monitor and assess running performance. A recent wearable system (i.e., Stryd™) calculates power production while running, separating this metric into two parts: power and form power. Apparently, power reflects the power output associated with changes in the athlete’s horizontal movement. Form power, however, represents the power output production caused by the combination of the oscillatory up and down movements of the centre of mass and lateral power when the athlete moves forward. This system employs mathematical calculations to estimate these two parameters from kinematic data collected from the described movements executed by the runner’s foot [16]. In addition, in a recent review [17], the Stryd foot pod was noted for its reliability and compatibility with metabolic power, compared to other commercially available portable running power devices.

While several studies have already analysed power output in running [18,19] and others have investigated the relation between VO2max and power production [16,20], to the best of the authors’ knowledge, there are no studies assessing the difference in power output between shod and barefoot running. In order to bridge this gap, this study aims to identify the effect of footwear on power output in endurance runners. It is hypothesised that increased effectiveness, leg stiffness and power production would be identified in barefoot running.

## 2. Materials and Methods

### 2.1. Subjects

Forty-one recreationally active endurance runners (25 males; age = 28.5 ± 6.9 years; height = 1.73 ± 0.08 m; body mass = 68.2 ± 11.6 kg), recruited by convenience, volunteered to take part in this study. All the participants were 18 years of age or older, capable of running 10,000 m in under 50 min (44.02 ± 4.22 min), injury-free for the last 6 months and were completing no fewer than 2 running sessions per week, therefore meeting the inclusion criteria. Every participant signed a formal consent form, aligned with the bioethics of the World Medical Association’s Declaration of Helsinki (2013). Once the objectives and procedures of the study were explained, participants were assured that they were free to leave the study at any time. The study was approved by the Ethics Committee of San Jorge University (009-18/19), from which sport sciences students were recruited. 

### 2.2. Procedures

Participants completed two testing trials over a one-session running protocol at their preferred comfortable velocity (11.71 ± 1.07 km·h^−1^) for data collection at the San Jorge University Biomechanics Laboratory (Zaragoza, Spain) in April 2019. Both trials were completed on a motorised treadmill with a slope maintained at 0% (HP cosmos Pulsar 4P; HP cosmos Sports & Medical, Gmbh, Nußdorf, Germany). Participants warmed up for 5 min on the treadmill where the velocity was increased and decreased several times until a comfortable velocity was achieved [21]. For each trial, participants completed two successive 3 min running bouts (i.e., shod for the first and barefoot for the latter), separated by a 2 min period to change from shod to barefoot condition. Since power output [19] and spatiotemporal parameters [22] reach a steady state in less than 2 min, data were recorded during both running trials and 6–8 strides were analysed [23].

### 2.3. Materials and Testing

Both body weight and height were measured for each participant, utilising a weighing scale (Tanita BC-601; TANITA Corp., Maeno-Cho, Itabashi-ku, Tokyo, Japan) and a stadiometer (SECA 222; SECA Corp., Hamburg, Germany), respectively.

For this study, a commercially available wearable power meter, Stryd™ (Stryd Summit Powermeter; Stryd, Inc., Boulder, CO, USA), was clipped on the laces of the runner’s shoe when running shod and placed and secured with tape on the runner’s instep during barefoot running (Figure 1). This lightweight, reinforced carbon-fibre foot pod (weight: 9.1 g) is based on a 6-axis inertial motion sensor (3-axis gyroscope, 3-axis accelerometer) and provides kinetic and kinematic data. During barefoot running, participants ran with socks to avoid friction injuries to the soles of their feet caused by the treadmill belt. When running shod, participants wore their traditional training shoes. The power meter was linked to the manufacturer’s mobile application (StrydApp, version 5.13), downloaded on a smartphone (iPhone 8, Apple Inc., Cupertino, CA, USA), for recording data.

Average power output (w; ratio of total of watts generated to total run time), form power (w; previously described), mean power output (w (MPO)), normalised MPO (w/kg (MPOnorm)), vertical oscillation (cm; quantity of up and down movement generated during running), leg stiffness (kN/m; ratio of the maximal force at the initial touchdown to the maximum leg compression at the middle of the stance phase) and running effectiveness (kg/N; ratio of speed to power) were obtained using the Stryd™ power meter.

Additionally, the running spatiotemporal parameters of contact time (time the foot spends in contact with the ground (CT)), flight time (time from toes-off to initial contact of the same foot (FT)), step length (distance covered between initial contact of one foot and the initial contact of the other foot (SL)) and step frequency (number of ground contacts that occurred in a minute (SF)) were also measured utilising the Stryd™ system, which has been previously validated for such purposes [18].

The foot strike pattern (FSP) exhibited by the participants was recorded using high-speed video at 240 Hz (Imaging Source DFK 33UX174, The Imaging Source Europe GmbH; Bremen, Germany). The camera was placed perpendicular to the treadmill from a sagittal view at 2 m from the centre of the treadmill and at a height of 0.30 m, which has been previously validated for such a purpose [24]. Three different FSP were identified in the present study [4]: rearfoot strike pattern (RFS), where the heel contacts the ground first; MFS, in which the outside edge of the foot contacts the ground first; and FFS, where the forefoot touches down first.

### 2.4. Statistical Analysis

Descriptive data are shown as mean (±SD), frequency and percentage. To determine the differences between nominal variables, McNemar’s test was used. The mean differences between values were analysed via pairwise mean comparisons (*t*-test) and the magnitude of the differences was expressed by means of the Cohen’s d effect size (ES) and interpreted as trivial (<0.19), small (0.2–0.49), medium (0.5–0.79) and large (≥0.8) [25]. All statistical analyses were performed using SPSS (version 25, SPSS Inc., Chicago, IL, USA) and statistical significance was accepted at α = 0.05.

## 3. Results

Significant differences (*p* < 0.05) were found in spatiotemporal gait characteristics during running when comparing shod and barefoot conditions (Table 1). When running barefoot, lower values for CT (*p* < 0.001, ES = 0.46) and SL (*p* = 0.003, ES = 0.13) were obtained with greater SF (*p* < 0.001, ES = 0.59), compared to those reported during shod running at the same comfortable velocity. The prevalence of FSP significantly differs (*p* < 0.034) between conditions, with 19.5% of runners showing RF, 56.1% MF and 24.4% FF during the shod condition, whereas no runners showed RF during barefoot running, with 31.8% and 68.2% showing MF and FF, respectively.

The comparisons between conditions (i.e., shod vs. barefoot) revealed no differences in MPO (*p* = 0.582, ES = 0.02) and MPOnorm (*p* = 0.568, ES = 0.03), whereas significant differences were found in form power, in both absolute (*p* = 0.001, ES = 0.14) and relative values (*p* < 0.001, ES = 0.33), running effectiveness (*p* = 0.006, ES = 0.36), stiffness (*p* = 0.002, ES = 0.20) and vertical oscillation (*p* < 0.001, ES = 0.48) (Table 2).

## 4. Discussion

This study sought to determine the effect of footwear on power output in long distance runners, comparing data collected by the Stryd™ system during both shod and barefoot running. The main finding of this study was that endurance runners showed greater running effectiveness when running barefoot in comparison with shod running, being consistent with lower values in form power and lower vertical oscillation. Additionally, our findings support those by previous studies reporting biomechanical alterations in runners who changed from traditional shoes to barefoot while running, such as the adoption of MFS or FFS, higher SF, and both shorter CT and SL [4,5,6,7]. Given the novelty of power output in endurance running, there is a lack of scientific evidence regarding this metric and, in particular, with power output in barefoot running, making this discussion challenging.

From a biomechanical standpoint, the tendency towards the runner’s adoption of MF or FF when switching from traditional running shoes to barefoot is supported by Lieberman and colleagues, as they stated that habitually shod runners adopted flatter foot positions at the initial contact when barefoot running [4]. In the same line, the runner’s showed significantly higher SF (3%) when running barefoot (166.99 ± 8.22 spm, *p* < 0.001) during the present study; this reinforces previous works [5,6], which found that habitually shod runners significantly increased their SF when barefoot running [5] and as velocity increased [6]. Likewise, Cochrum and colleagues reported that barefoot running at 50% VO2max resulted in a 2.4% shorter SL in comparison with shod running conditions [6], endorsing the findings reported here, where SL is significantly shorter in barefoot running (1.83%) in comparison with running in shod conditions (1.09 ± 0.15 m and 1.11 ± 0.15 m, respectively; *p* < 0.05) at a comfortable velocity.

Significantly smaller CT was also reported in barefoot running (0.252 ± 0.019 s, *p* < 0.001), supporting previous studies [5,7]. Divert and colleagues [5] reported that CT significantly increased in shod running, and Lussiana and colleagues [7] found shorter CT when comparing minimalist to traditional shoes. Although all these studies share shod–barefoot comparison in their procedures, the slight differences may be due to their different methodologies. While our participants completed one single, steady-state, comfortable velocity testing session on a motorised treadmill at a maintained slope of 0%, the studies discussed above are based on an incremental gradient protocol [7], a two-session protocol made up of six running bouts of 4 min [5], and four treadmill running testing sessions [6]. It should also be noted that none of those studies utilised the Stryd™ power meter to analyse the spatiotemporal parameters, in either the shod or barefoot condition.

Considering that it has been proposed that the addition of 100 g to the foot reduces running economy by 1% [5,26], barefoot running would optimise the stretch-shortening cycle behaviour, buffering and releasing elastic energy [27] and increasing leg stiffness by the adoption of a plyometric movement pattern [28]. Moreover, as the foot’s core muscle system produces an adaptation during compression of the longitudinal arch, which results in increased ability to recoil elastic energy over the stance phase [10,11], one might expect greater power output in barefoot running compared to running in the shod condition. It should be noted that the footwear condition does not influence MPO or MPOnorm, but it does significantly influence form power (watts and percent), running effectiveness, leg stiffness and vertical oscillation in endurance runners.

Given that power can be defined as the product of force and velocity [29], and that in the present study both running bouts (i.e., shod and barefoot) were executed at the same comfortable velocity for every participant, it seems reasonable that MPO and MPOnorm remained stable under both footwear conditions. The power output values reported in the present study during shod running (210.05 ± 44.16 W) are supported by those found by previous works using the Stryd™ power meter at the same running velocity [15,19].

From a practical application of MPO in endurance runners, the Functional Threshold Power is a performance index referring to the highest MPO maintained for around 60 min running without the onset of fatigue [30], commonly used to determine training intensities (i.e., training zones) and quantify athletes’ responses to training stimuli [30,31]. In a recent study, MPO and MPOnorm have shown a strong relationship with the Functional Threshold Power at submaximal running from 10 min to 30 min [32]. Our results have shown no significant differences for MPO and MPOnorm between the shod and barefoot running conditions, suggesting no changes in load intensities between the two footwear conditions. Thus, a runner could maintain their training loads based on watts. However, the authors recommend being cautious with this information as intra-articular loads could be different due to biomechanical changes caused by barefoot running, such as the change in FSP from RFS to FFS, greater ankle stiffness, lower impact load and brake load, greater knee flexion at ground contact, and reduced tibialis anterior muscle activity, among other factors [33].

Regarding form power, and according to the manufacturer’s manual (https://www.stryd.com/guide (accessed on 1 June 2022)), most athletes would exhibit form power ranging from 30 to 100 W. The findings reported in our study show significant differences between both running conditions (69.95 ± 12.51 W when shod, 68.28 ± 12.19 W when barefoot) and seem to align with the manufacturer’s statement. It can be argued that form power relates to leg stiffness and vertical oscillation in endurance running in different ways. It is known that lower-limb stiffness varies across footwear conditions, resulting in increased leg stiffness when barefoot in comparison to shod running [5,7,8,10,27]. This statement is supported in the present study as the leg stiffness values are significantly greater in barefoot running (10.65 ± 1.93 kN/m). Since increased leg stiffness optimizes elastic energy recoil and enhances running economy [34], it is reasonable to find lower values of form power and increased leg stiffness in barefoot running, making the reverse equally valid in shod running. The values for vertical oscillation exhibit significant differences between shod (7.93 ± 0.98 cm) and barefoot running (7.48 ± 0.9 cm). The significantly lower values found in barefoot running demonstrate that runners show less vertical oscillation as they run, positively affecting running economy [35,36], which is also associated with the increased leg stiffness found under this footwear condition [7], consequently producing lower form power under this footwear condition.

The Stryd™ system also offers a running effectiveness metric. This novel metric is referred to as the ratio of running velocity to power (https://www.trainingpeaks.com/blog/wko4-new-metrics-for-running-with-power/ (accessed on 1 June 2022)). The findings reported here align with the proposed values for this value (i.e., ~1 kg/N), showing significantly higher effectiveness (*p* < 0.005) in barefoot running (0.97 ± 0.06 kg/N). Although it has not been reported before, this metric might be useful for coaches and practitioners as white papers have stated that the closer the running effectiveness value to 1 kg/N, the more effective runners are at transforming external power into velocity (https://docs.google.com/document/u/2/d/e/2PACX-1vTzjH-Ns_GInUm4lAxi3cVOQpzzKcWNF6VEX271s-QGYFHjwMgyLhhmu5i21-1_CaC3eL0B817rQo8k/pub (accessed on 1 June 2022)). This metric must not be used interchangeably with running economy as they are completely different parameters. However, following the statements of the aforementioned white papers, running effectiveness might represent running economy from a mechanical standpoint.

Regarding running economy, referred to as the energy required to maintain submaximal velocity efforts [37], several studies have not found significant differences between shod and barefoot running [38,39]. These studies did not control FSP, which may influence these comparisons [5]. Of note, after controlling for maximal oxygen consumption (VO2max) and footwear conditions (i.e., barefoot, minimal, and traditional running shoes), Cochrum et al. [6] stated that barefoot running provides less metabolic benefit over cushioned shoes. This finding is supported by previous work, whose authors suggested that the design of cushioned shoes offers metabolic savings compared to barefoot running [28]. Our results, in contrast, provide evidence that barefoot running can be more metabolically beneficial than running in shoes. In addition, we believe that the transition from running in traditional shoes to less cushioned shoes, minimalist or barefoot running should be carried out gradually, as has already been recently proposed in a 10-week pain and injury free retraining program [40].

The findings described here are based on entirely mechanical parameters; therefore, they should not be transferred to physiological terms. These findings should also not be extrapolated to injury management or competition, as we detailed the changes that occur in shod and barefoot running regarding kinetic and kinematics parameters. 

Finally, there are some limitations to consider. Firstly, the protocol was completed on a motorised treadmill at a comfortable velocity, preventing the readers from extrapolating these findings to other velocities. The participants wore their own running shoes in shod running, therefore increasing the ecological validity of the study. It would be of interest for the research community to assess running power output and related metrics considering different types of footwear, given the current revolution in the design of running shoes. The participants were habitually shod runners; therefore, the novelty of the task might influence the outcomes of barefoot running. Ultimately, the lack of scientific evidence on this topic made the discussion section especially complex. However, notwithstanding the aforementioned limitations, the present study offers new insights into the power production in endurance running, as well as the use of power meters and the interpretation of the metrics they provide, which might be of high value for clinicians, coaches and athletes who aim to introduce this metric into training and competition.

## 5. Conclusions

The results obtained show that, besides the already known spatiotemporal gait characteristic adaptations, barefoot running reported greater values in running effectiveness in comparison with shod running, being consistent with lower values in form power and lower vertical oscillation related to running economics. Future studies are needed to examine whether the long-term effect of short periods of barefoot running might contribute to increased running efficiency and leg stiffness in endurance runners, which would affect running economy.

## Figures and Tables

**Figure 1 sensors-22-04828-f001:**
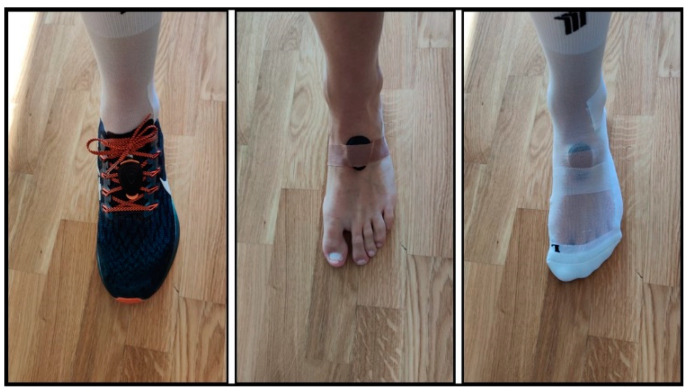
Representation for the placement of the Stryd™ power meter clipped on the laces of the runner’s shoe (left picture) and placed and secured with tape on the runner’s instep during barefoot running (central and right picture).

**Table 1 sensors-22-04828-t001:** Spatiotemporal gait characteristics during running shod and barefoot at comfortable velocity.

		Shod Condition	Barefoot Condition	*p*-Value (d)
FSP (n, %) ^	RF	8 (19.5)	0 (0)	0.019
	MF	23 (56.1)	13 (31.8)	0.033
	FF	10 (24.4)	28 (68.2)	0.012
CT (s)		0.261 (0.020)	0.252 (0.019)	<0.001 (0.46)
FT (s)		0.111 (0.018)	0.108 (0.017)	0.053 (0.17)
SL (m)		1.11 (0.15)	1.09 (0.15)	0.003 (0.13)
SF (spm)		162.06 (8.06)	166.99 (8.22)	<0.001 (0.59)

^ indicates that a McNemar test was conducted to compare frequencies; d: Cohen’s d effect size; FSP: foot strike pattern; RF: rearfoot; MF: midfoot; FF: forefoot; CT: ground contact time; FT: flight time; SL: step length; SF: step frequency.

**Table 2 sensors-22-04828-t002:** Power output and related parameters during running shod and barefoot at comfortable velocity.

	Shod Condition	Barefoot Condition	*p*-Value (d)
MPO (W)	210.05 (44.16)	210.73 (44.24)	0.582 (0.02)
MPO_norm_ (W/kg)	3.07 (0.32)	3.08 (0.32)	0.568 (0.03)
Form power (W)	69.95 (12.51)	68.28 (12.19)	0.001 (0.14)
Form power (%)	33.6 (2.8)	32.7 (2.7)	<0.001 (0.33)
Running effectiveness	0.95 (0.05)	0.97 (0.06)	0.006 (0.36)
Leg Stiffness (kN/m)	10.26 (1.86)	10.65 (1.93)	0.002 (0.20)
Vertical oscillation (cm)	7.93 (0.98)	7.48 (0.90)	<0.001 (0.48)

d: Cohen’s d effect size; MPO: mean power output; MPO_norm_: normalised mean power output.

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
