# Peer review of "Influence of the Shod Condition on Running Power Output: An Analysis in Recreationally Active Endurance Runners"

_sensors, 2022, doi:10.3390/s22134828_

Round 1
Reviewer 1 Report
INTRODUCTION
line 35: "jeopardising, therefore, running economy". The meaning of this sentence is not clear and it is not necessary to include it in the introduction section.
-The contributions in lines 52 and 53 are more appropriate in the discussion section.
MATERIAL AND METHOD
-Delete "1" in line 85.
-For better understanding describe the STRYD sensor placement (lines 118-122) before the variables (lines 110-117).
DISCUSSION
Although well structured, it is too long. The findings in the results are well discussed but it is unnecessary to determine why there are discrepancies with the results of other studies.
CONCLUSION
A series of recommendations or reflections are established that do not belong in a conclusions section. Up to line 283 the conclusions are correct. From that line on, they are reflections from the authors.
Reviewer 2 Report
sensors-1778072 _review
Title: Influence of the shod condition on running power output: An analysis in recreationally active endurance runners
Comments and Suggestions for Authors
Dear authors,
I was glad to have the opportunity to review this manuscript that aims to identify the effect of footwear on power output in endurance runners. As has been well described in the article, the present study offers new insights into the power production in endurance running as well as the use of power meters and the interpretation of the metrics they provide, which could be of great value both for training and in competition as it would provide very interesting data to runners and their coaches. You concluded that running barefoot showed a greater running effectiveness in comparison with shod running, being consistent with lower values in form power and lower vertical oscillation. In my opinion, therefore, further researches are needed in this field in order to analyse whether the long-term effect of barefoot running exercises could lead to increased running efficiency and leg stiffness in endurance runners, and how this might affect running economy.
In my opinion, the introduction, materials and methods, results and discussion are clearly described. In general, the article is well-written and it is easy to follow authors’ thoughts and reasoning
In general, the manuscript is well-written and the text is understandable and organized.
I would like to comment on some minor issues that could be addressed to improve the document, in my opinion.
Specific comments:
Abstract:
- Page 1, line 16: In this line appears: Mean power output (MPO and MPOnorm). What do the acronyms MOPnorm stand for? Please, add this information.
Introduction
- Page 1, lines 27-33.
The term “endurance runners” is central to your manuscript. I think it would be interesting to add more information about it. Add a reference if necessary.
Material and methods
- Page 3, lines 126-128. You mentioned that the Institutional Review Board approved this study. Please add more information if possible in this section, i.e. code, identification number….
- Page 3, lines 96-97. Could you add information about the period of time in which the data were recorded and the university where this study was carried out (country)?
-Page 3, line 115-117: Please report the reliability and validity for the Stryd Summit Powermeter if possible, add some reference that supports it, if necessary.
- Page 3 lines 118-119. In the limitations of the study you mentioned that: “The participants wore their own running shoes in shod running”. You describe well how the barefoot race was carried out; however, you do not mention anything about the type of footwear that the runners wore to carry out the shoe race. I suggest that you add this information also in this section, I think it is essential for your work I suggest you add this information also in this section, I think it is essential for your work.
- Page 4, line 138-144. There is no statement about the sample size. How were the sample size calculated? This information is very important to support the external validity of the study findings. Please add this information.
Results
- The results section is well-structured and comprehensive.
Discussion
-Your discussion section is in general adequate and complete.
In the introduction section (page 2, lines 55-56) you mentioned that previous studies report that: “barefoot running provides less metabolic benefit over cushioned shoes”. I consider that this idea can be very interesting and I do not see it reflected in your discussion. It is true that since there is not much previous bibliography, it can be difficult to carry out a discussion. But it is also interesting and should be included if there are any, as in this case, comments on articles that contradict your results. I suggest you add this information in this section.
Conclusions
- Page 7, lines 286-288. I suggest rephrased this sentence in a more careful way. i.e….Future studies are needed to examine in depth and in a larger number of runners whether the long-term effect of short periods of barefoot running could contribute to increased running efficiency and leg stiffness in endurance runners, which It would affect the economy of the race.
I hope that my comments could help to improve the paper.
Congratulations for your research.
